# Crossover of Work Engagement: The Moderating Role of Agreeableness

**DOI:** 10.3390/ijerph19137622

**Published:** 2022-06-22

**Authors:** Konstantinos Chiotis, George Michaelides

**Affiliations:** 1Department of Psychology, Aristotle University of Thessaloniki, 541 24 Thessaloniki, Greece; 2Employment Systems and Institutions, Norwich Business School, University of East Anglia, Norwich NR4 7TJ, UK; g.michaelides@uea.ac.uk

**Keywords:** work engagement, crossover, agreeableness, dyad, intervention

## Abstract

Work engagement can cross over from one individual to another, and this process may depend on several factors, such as the work context or individual differences. With this study, we argue that agreeableness, one of the Big five personality measures that characterized empathetic, can be instrumental in the crossover process. Specifically, we hypothesize that agreeableness can facilitate this process so that engagement of an actor can more easily cross over to their partner when either of them or both have high agreeableness. To evaluate our hypotheses, we implemented an intervention to the working schedules of 74 participants for two weeks. The intervention involved pairing participants to work together so that to create dyads with varying levels of dissimilarity. The results from a multilevel regression model indicate that there is a crossover effect and partner’s work engagement can be transferred to actor after a two-week collaboration. This effect is further intensified if either one or both members in the dyad are characterized by high levels of agreeableness. These findings help to decode the mechanisms underlying the crossover process and illustrate how to ideally coordinate work dyads to take advantage of the crossover effect and maximize employee engagement.

## 1. Introduction

Crossover is a form of emotional contagion, and a conscious procedure in which transference of emotions and affective states is facilitated by the empathic reactions of partners [1]. Simply put, when we spend time with others, and pay close attention to them, our affective states can cross over. Crossover processes can cause affective states, emotions, or well-being to be transmitted between colleagues [2,3], leaders and followers [4,5,6,7], and dual-earner couples [8,9,10]. Work engagement, in particular, has received extra attention within the crossover literature. Engaged employees are enthusiastic about their jobs [2] and can be a source of inspiration for others [3]. Research has shown that work engagement can be transmitted among working spouses [11,12], as well as from one employee to another in the workplace [2,3,7]. This is very important, considering the various positive outcomes of work engagement at the individual, team and organizational levels. For example, due to their dedication, engaged employees demonstrate better in role task performance [13], have better financial returns [3], and improved well-being [14]. In addition, engaged employees, due to their physical, cognitive, and emotional connection to their work roles [15] are more likely to be entrepreneurial [16] and exhibit proactive behaviors [17]. The fact that engagement can cross over from one employee to another, means that it can also emerge as a collective characteristic of work teams [2,18]. Indeed, there are studies that have highlighted the positive association between team level work engagement and team performance [19,20] as well as the positive relationship between work engagement and performance at the organizational level [21,22].

Past research has shown that the transmission of work engagement can be facilitated by various factors [23,24]. These may include factors such as frequency of communication e.g., [3] or more individual characteristics, such as empathy [25,26]. This brings into question whether broader personality traits, such as those measured by the Big-5, can also influence the extent to which engagement can be transmitted between individuals.

To better understand the role of personality in the crossover process, in this paper, we focus on agreeableness as a key catalyst for the crossover process, and we examine whether it moderates the crossover of work engagement from one employee to another in the workplace. Agreeableness is one of the main personality dimensions of the Big-5 model of personality and characterizes individuals who are cooperative, empathetic, and altruistic, and typically engage in prosocial behaviors [27]. We evaluate the role of agreeableness in this process, in an intervention study designed to facilitate the crossover of engagement in nursing staff of a public hospital. The intervention involved the manipulation of the work schedule of nurses for two weeks, as to form dyads with varying levels of engagement dissimilarity.

The importance of this research is three-fold. First, by investigating agreeableness as a moderator in the crossover of work engagement, we add a new perspective to the crossover literature. Given the higher levels of interdependence that employees may experience due to increased demands for working in teams or pairs [28], as well as the positive organizational outcomes of work engagement, it is crucial to investigate the effect of employees’ personality on the transmission of this positive affective state. Second, we aim to evaluate and extend previous relevant findings in the crossover process among employees e.g., [2,3,7] by conducting a study in a rather demanding workplace, such as a hospital. This will make an important contribution to the evidence base regarding the crossover of positive experiences among employees. Finally, our study is the first to examine crossover of work engagement through a methodological design based on an intervention, with data collected from the participants on two different occasions: before (T1) and after (T2) an intervention. Specifically, the intervention concerns the formation of working dyads with employees that are characterized by different initial (T1) levels of work engagement, and the collaboration of these dyads for a period of two weeks. The ecological validity afforded by this design strengthens both the utility of the results and their application to real work environments for designing interventions to improve work engagement.

### 1.1. Work Engagement

Kahn [15] was the first that introduced the construct of personal engagement at work, defining it as a psychological, emotional and physical state transferring peoples’ energy into their works. According to this view, an authentic expression of self occurs during situations of engagement, which in turn is beneficial for the individual. Regarding work engagement, there is a broad consensus among scholars about its core dimensions which are energy and involvement [29] and Kahn’s [10] conceptualization of engagement suggests that it entails a behavioral-energetic, an emotional, and a cognitive component [30]. From this perspective, Schaufeli and Bakker [31] define work engagement as a positive, fulfilling, and work-related state of mind, characterized by vigor, dedication (emotional component) and absorption (cognitive component). Vigor refers to having high levels of energy and mental resilience while working (behavioral-energetic component). Dedication refers to a sense of enthusiasm, pride, and significance that someone feels due to his/her work (emotional component), and finally absorption refers to being highly concentrated to work or a specific task so that it can even become rather difficult for a person to detach himself/herself from work (cognitive component).

Work engagement has been found to relate to several positive organizational outcomes including organizational commitment and improved well-being [14,31], job performance [3,32], and lower turnover intention [33]. Engaged employees are quite energetic, self-efficacious [34], willing and happy to assist their colleagues [35], they tend to create their own positive feedback, and despite committing a lot of effort and resources, they are still left with a state of fulfillment and satisfaction about what they have accomplished [29].

### 1.2. Crossover of Work Engagement

Crossover is defined as the process that occurs when psychological well-being experienced by a person affects the level of psychological well-being of another person in the same social environment [36]. Consequently, crossover is a dyadic inter-individual transmission of mental states or emotions among closely related individuals, which occurs within a particular domain of life such as family or workplace [37].

Westman and Vinokur [38] proposed three main mechanisms responsible for the crossover process. The first mechanism refers to a direct transmission of from one partner to another through empathetic reactions (direct empathetic crossover). They argue that well-being experienced by one partner produces an empathic reaction in the other, which in turn leads to an increase in his/her own level of strain. The second mechanism concerns an indirect transmission of well-being as a result of interpersonal transactions and behavioral interactions between partners (indirect crossover). Thus, experiencing some change in well-being by one person can trigger a change in their behaviors as well as the way they interact with others and consequently influence their well-being. As such, social undermining or conflicting interactions could be mediating the crossover of negative affective states, and social support can mediate the crossover for positive affect [12]. Finally, the third mechanism suggests that well-being experienced by partners is not actually due to any crossover effect, but to common stressors and shared contexts which affect both partners. These three mechanisms can either operate independently of one another or jointly [39], and findings of empirical studies [25,40,41], support all of these propositions.

Although Westman [36] initially placed the emphasis on negative forms of well-being, such as job stress, strain, and burnout, it is possible that positive experiences may also cross over from one partner to the other via the same mechanisms as the negative aspects [36]. Work engagement is one of the most widely researched positive aspects of individual well-being [42]. In terms of its position in the nomological network of well-being constructs, it is negatively related to job burnout [43], and positively with life satisfaction [44,45], and happiness [46]. Moreover, when employees experience work engagement they also have high levels of intrinsic work motivation [47] which is to say that they tend to find enjoyment in the job itself regardless of whether there are any additional rewards or benefits associated with doing it [47].

Based on the research of emotional contagion in the workplace e.g., [48,49,50,51], there is a considerable amount of studies that have focused on the crossover of work engagement among employees and have also attempted to uncover the factors and the conditions that moderate this process. For example, Wirtz et al. [7] study showed that work engagement can cross over from subordinates to their leaders over time, indicating this way that subordinates can shape their leaders’ work experiences and affect their well-being. In their diary study, Bakker and Xanthopoulou [3] showed that daily work engagement crossed over from one employee to the other within a dyad, only on those days that these employees interacted more frequently than usual, indicating that frequency of communication played a moderating role in the transmission of work engagement. In another study, Bakker et al. [2], found that crossover of work engagement, and especially vigor, took place on days when colleagues interacted more frequently, and that expressiveness built through frequent daily interaction, could increase the possibility for work engagement to cross over from one employee to another. Similarly, Tian et al. [52] showed that work engagement crossed over from one partner to another and communication quality had a significant moderating effect on this process. In formulating our hypotheses, we consider the partner to be the recipient in the crossover process. We thus hypothesize that:

**Hypothesis 1 (H1).** *An actor’s initial (T1) work engagement, will be positively related to their partner’s work engagement after a two-week collaboration (T2)*.

### 1.3. The Relationship between Work Engagement and Agreeableness

Agreeableness is one of the major personality dimensions of the five-factor model (Big 5) of personality [53,54]. It is a super-ordinate personality characteristic consisting of lower-level traits or facets such as altruism, straightforwardness, trust, modesty, compliance and tender-mindedness [55]. Thus, agreeableness captures the degree to which individuals are cooperative with others and agreeable individuals are characterized as kind, sympathetic and considerate [27,56].

Agreeable individuals are more likely to foster strong work relationships with colleagues, which can have positive effects on work engagement. This is because agreeable colleagues can provide support, which has been consistently shown to have a positive effect on work engagement [37,57]. Moreover, establishing strong relationships helps to fulfill the need for relatedness which is one of the basic psychological needs and a pre-cursor of motivation and work engagement [58,59]. Agreeableness is also closely related to prosocial behavior [60,61] which refers to the tendency to help, donate, comfort and care for others [62]. As such, we can expect that individuals who are agreeable are more likely to directly want to influence the well-being of others.

From an empirical perspective, numerous studies have offered support to a positive association between agreeableness and work engagement. In a recent meta-analysis across a total of 30 samples, Young et al. [63] found a modest positive correlation between agreeableness and work engagement. More specifically, in terms of health care professionals, Scheepers et al. [64], found that agreeableness was positively related to clinician supervisors’ work engagement and Perez-Fuentes et al. [65] found the same with a large sample of nurses. We thus, hypothesize:

**Hypothesis** **2a** **(H2a).**
*Partners’ agreeableness is positively related to their work engagement at T2.*


From a similar perspective, being on the receiving end of agreeableness can also have a positive effect on employees’ engagement. As agreeableness implies, being more sensitive to other’s needs, can potentially be an additional source of support or provide other resources which can increase work engagement. Indeed, according to the Job Demands-Resources model [66], co-worker support is a job resource that may enhance work engagement [67]. In their longitudinal study, Schaufeli et al. [68] found that an increase in job resources (social support among them) predicts future work engagement. Based on this, we can hypothesize that,

**Hypothesis** **2b** **(H2b).**
*An actor’s agreeableness is positively related to their partner’s work engagement at T2.*


### 1.4. Agreeableness as a Crossover Booster

Further to the direct effects of agreeableness on engagement, we can also expect agreeableness to enhance and further facilitate the crossover process. From the direct crossover perspective, agreeableness can enable individuals to exert a stronger direct empathic reaction to others and at the same time be more empathetic receivers themselves.

Agreeableness is also strongly related to empathy [56,69,70] which is a key ingredient for direct empathic reaction [3,36,71]. For example, Bakker and Demerouti [25], showed that perspective taking—the ability to understand or perceive a situation from the perspective of another person [72]—moderated the crossover process of work engagement between working couples. However, they also showed that empathic concern did not have such an effect. In a similar study, Bakker, Shimazu et al. [26] found that perspective taking moderated crossover of work engagement between Japanese couples, but only when women were the receivers of the information. Moreover, this study showed that engagement crossover was even more pronounced when both men and women were high in perspective taking, as it enabled individuals to better communicate their affective states and influence each other. Similarly, we would expect that agreeableness of both members of a work dyad would be instrumental in facilitating the crossover process.

Going beyond the direct empathic reaction, agreeableness can also enhance the indirect crossover mechanism. When experiencing a positive affective state people are more likely to be co-operative, supportive, and sympathetic to their colleagues and through such interactions can instigate experiences of positive affective states. However, since being co-operative, supportive, and sympathetic to others is the essence of being agreeable, we can expect that agreeableness can amplify this process. Combining the above mechanisms, we hypothesize that agreeableness of either the actor or the partner will augment the crossover process.

**Hypothesis** **3a** **(H3a).**
*The positive relationship between an actor’s work engagement at T1 and their partner’s work engagement at T2 is stronger when the partner’s agreeableness is higher.*


**Hypothesis** **3b** **(H3b).**
*The positive relationship between an actor’s work engagement at T1 and their partner’s work engagement at T2 is stronger when the actor’s agreeableness is high.*


We show in Figure 1 a diagrammatic representation of the model that combines all of our hypotheses.

## 2. Materials and Methods

### 2.1. Participants

The sample of the study consisted of nurses working in a Greek military hospital. Initially, 90 nurses were invited to the study and 74 of them agreed to participate (response rate 82.2%). The study was approved by the scientific council of the hospital. The sample consisted of 59.5% female participants, the mean age was 35.37 years (SD = 7.61), and the mean tenure was 13.27 years (SD = 7.21). 14.9% of the participants were high school graduates, whereas the rest were graduates of higher educational institutions.

### 2.2. Measures

*Work engagement* was measured with the Utrecht Work Engagement Scale—UWES-17 [73]. This scale consists of 17 items (UWES-17) which measure three dimensions of work engagement, vigor (VI), dedication (DE) and absorption (AB), as described by Schaufeli et al. [43]. Items were measured with Likert scales ranging between 0 (never) and 6 (every day). UWES was designed for the assessment of these three dimensions, as well as for the overall work engagement [74,75]. According to Schaufeli and Bakker [67], the assessment of the overall work engagement could be more useful than the evaluation of its dimensions separately. Indeed, in our sample the correlations were very high (all above 0.7) which is an indicator of potential multicolinearity and using them separately would be redundant. Chronbach’s alpha for the combined scale was 0.96 for both time 1 and time 2 measures

*Agreeableness* was evaluated using the Traits Personality Questionnaire (TPQue) [76]. The TPQue is based on Costa and McCrae’s definitions of the Big five model (extraversion, neuroticism, openness to experience, agreeableness, and conscientiousness), adapted to the Greek ethnic and cultural characteristics of the population [76]. Each factor consists of 6 sub-factors, and each sub-factor consists of 6 items, thus there are 36 items for each of the five personality factors. Each of these items is scored through a Likert scale that ranges between 1 (I totally disagree) and 5 (I totally agree). Chronbach’s alpha for agreeableness using the 36 items was 0.83.

*Control variables* included age and gender at the individual level, and whether the two members of the dyad were of the same or different genders. Furthermore, we also controlled for work engagement at time 1. By doing so, we are removing the pre-intervention part of the variance from the engagement variable and model the residual change in work engagement. However, except for work engagement at time 1, the rest of the control variables were not significant, and we therefore decided to exclude them from the final analysis.

### 2.3. Procedure

Participants were asked to fill in a general questionnaire to collect demographic information, and then complete questionnaires measuring their individual personality characteristics and the level of their work engagement respectively. The questionnaires included an ID code number to allow collating the data whilst protecting anonymity of the participants. Engagement data were collected before and after a two-week intervention (T1 and T2), whilst agreeableness was only measured once at the beginning of the study (T1).

The intervention consisted of formulating work dyads or partnerships between nurses according to their levels of engagement. This was achieved with the co-operation of the managers of the various medical departments. Based on initial (T1) levels of engagement, we formulated a variety of combinations as to maximize variability. Specifically, we formulated 26 dyads that had high engagement dissimilarity and 11 dyads with low engagement dissimilarity. All dyads worked together for a period of two weeks in eight-hour shifts (14 working days), after which the participants were asked to answer again the UWES-17 questionnaire (T2). During these two weeks, dyads were allocated to accomplish typical tasks based on the specific features and needs of each medical department. Such tasks included, among others, recording patients’ vital signs, provision of medication to the patients, and preparing paperwork. Each of these tasks was performed by a dyad during the 8-h shift. We should mention that some participants may have worked with their partners in dyads before the implementation of our intervention, but such collaborations occurred rarely and surely not for 14 consecutive working days, preventing that way the establishment of a closer working relationship.

Since dyads were assembled based on the smooth and proper functioning of each medical department, we could not have a proper control group. For example, creating control groups that would be allocated to work together but would not really spend any actual time together would not be meaningful in this context. Similarly, and to ensure the continuous proper operation of the hospital, we could not have arranged for nurses to work on their own.

### 2.4. Analysis

Data were analyzed with multilevel models, which essentially requires allowing for an additional random intercept at the dyad level to account for the independence in engagement between members of the same dyad [77]. The independent variables were standardized to help with the interpretation of the results and were added to the model in three steps. In the first step, we evaluated H1: there is a main effect of partner’s engagement at time 1 on actor’s engagement at time 2. In the second step we evaluated the main effects of agreeableness (H2a) and agreeableness of partner (H2b), and in the third step we added the interactions between engagement of partner and agreeableness (H3a) and engagement of partner and agreeableness of partner (H3b). The analysis was conducted using R.4.1.0 [78]. At each step, we evaluated the improvement in model fit using a log likelihood difference test (χ^2^).

## 3. Results

Table 1 shows descriptive statistics and correlations between all variables used in the analysis. Note that means and standard deviations for partner and actor variables are the same. This is because everyone in the sample is both an actor and a partner, and thus the partner’s data are the same as the actor’s data.

As expected, there is a very strong correlation between engagement before and after the intervention, but almost no relationship with the engagement of the partner. This is due to the way we designed the intervention to ensure maximum variability in terms of the possible combination of dyads, which is also reflected in the negative correlation between actor’s and their partner’s engagement at time 1. However, it should be noted that this is also what we should expect if we had a very large number of randomly allocated dyads that have never worked with each other before.

Table 2 below shows results from the multilevel regression of the three steps. After controlling for engagement at time 1, there was significant engagement crossover (step 1: β = 0.22, *p* < 0.001, 95% CI = [0.16, 0.28]) providing support for H1. The model at step 1 was significantly different from the model that only had the control variable of baseline engagement. Interestingly, the correlation (Table 1) between the actor at time 1 and their partner at time 2 is non-significant and yet when tested via regression is positive and significant. The key difference between the two tests is that in the regression model we are controlling for the baseline levels of engagement at time 1. This suggests that without controlling for the baseline, there is a suppression effect masking the association of the actor’s engagement from time 1 to the partner’s engagement at time 2. Most likely, this effect is due to the strong association between engagement at time 1 and time 2 and the way we matched the dyads so to ensure dissimilar initial engagement. It is therefore only when we control for engagement at time 1 that we can see the true crossover effect.

At step 2, we included the two agreeableness variables (actor and partner) but there were no significant effects for either the partner’s agreeableness (step 2: β = 0.04, *p* < 0.05, 95% CI = [−0.02, 0.09]) or the actor’s (step 2: β = 0.05, *p* < 0.05, 95% CI = [−0.01, 0.10]) on engagement change. The model with the two agreeableness variables was not significantly better from the model in step 1. Thus, neither H2a nor H2b were supported.

At the final step, we tested the interaction effects between an actor’s engagement and agreeableness on their partner’s engagement. For H3a the effect suggests that the partner’s own agreeableness can moderate the crossover effect (β = 0.06, *p* < 0.01, 95% CI = [0.01, 0.11]). Similarly, for H3b this suggests that the actor’s agreeableness moderates the effect of their engagement on their partner’s change in engagement (β = 0.07, *p* < 0.01, 95% CI = [0.02, 0.11]). The final model was significantly different from the step 2 model which did not include the interaction effects. Including the interactions, the model explained 19% of the variance at the within level was and 40% at the between level.

To interpret the interactions, we plotted separate regression slopes at one standard deviation above and below the mean, and at the mean of engagement at time 1. As shown in Figure 2 and Figure 3 the crossover effect is stronger for high agreeableness and this result is consistent for both actors’ and partners’ agreeableness. Thus, both H3a and H3b were supported.

## 4. Discussion

In this study, we examined the crossover effect of work engagement between nurses in a military hospital and the potential moderating effect of agreeableness in this process. To evaluate our hypotheses, we designed an intervention in the working schedules of the participants, whereby they were paired to work in dyads for two weeks. From our analysis of the data, we found support for three of the five hypotheses (H1, H3a, and H3b).

Supporting the first hypothesis and consistent with previous crossover research, we found that work engagement crosses over from one partner to the other, after collaborating for a particular amount of time [3,25,26]. For our second set of hypotheses, however, we found that neither the actor’s nor the partner’s engagement had a main effect on the partner’s engagement. We expected that agreeableness would have these direct effects by virtue of increasing prosocial behaviors, social support, and relatedness, but it does seem that this is not the case. Since in our model we are controlling for the initial baseline levels of engagement before the intervention, the result reflects that agreeableness did not cause a change in engagement levels, even if it is associated with engagement. This is evident from the fact that agreeableness was significantly correlated with engagement at both before and after the intervention. This relationship is consistent with meta-analytical evidence [65] as we as other studies that examined agreeableness and engagement in healthcare professionals [66,67].

Finally, we did find support for the third set of hypotheses that the crossover effect is moderated by agreeableness for both members in the dyad. Specifically, crossover of engagement was found to be stronger when either the actor or the partner had high agreeableness. These results complement past research that identified similar moderators of the crossover process such as perspective taking of either the actor or the partner [25,26] and communication quality in dual-earner couples [52].

### 4.1. Implications for Theory and Practice

This study contributes to the crossover literature in several ways. First, it contributes to the knowledge base of how personality may facilitate the crossover process. To our knowledge, this is the first study to examine a Big-Five personality trait as a potential moderator of the crossover process. We investigate agreeableness as it closely related to prosocial behavior, which represents a broad category of acts that are beneficial to other peoples’ well-being [62], and potentially responsible for the transmission of positive affective states from one person to another. Agreeableness is also related to empathy, which has been shown to moderate crossover of work engagement between dual-earner couples [20,21].

Second, our results offer support for both the direct empathic crossover process and the indirect one [36,38,71] which in practical terms may not be two different processes, but two mechanisms of the same process that occur simultaneously. As our results indicate, it is the interaction between the two processes that determines the magnitude of the crossover of engagement, a fact that offers support to Westman’s [39] argument that the different crossover mechanisms can either operate independently of one another or jointly as well.

Third, whilst this is not the first study to examine crossover effects of engagement, it is, to our knowledge, the first to show this via a methodological design that includes an intervention. As Biron and Karanika-Murray [79] explain, one of the mechanisms through which interventions achieve their intended outcomes is emotional contagion. By designing an intervention that explicitly aims to maximize crossover and emotional contagion, we provide a clear direction for how managers and organizations can take advantage of this process.

From a practical point of view, our findings can be invaluable to organizations, who may be interested in implementing interventions to increase their employees’ work engagement. By considering the individual personality characteristics and the levels of work engagement of each employee, managers can strategically set up work schedules to match individuals in a way that maximizes crossover of work engagement. Interventions that develop job and personal resources can be effective tools in promoting work engagement [80,81] and matching individuals according to their agreeableness could potentially be instrumental is fostering resources such as social support from colleagues. Moreover, recruiters can take advantage of the evidence in this study and place more emphasis, within the selection process, on the measurement of agreeableness for jobs that require close collaboration between employees.

### 4.2. Limitations and Future Directions

The main strength of this research has been in answering the research questions via a quasi-experimental intervention study. Specifically, we manipulated the working schedule of nursing staff to formulate work dyads according to the initial levels of engagement of participants. This is a unique research design and, to our knowledge, the first time that such an intervention has been applies to understand crossover effects.

Nevertheless, the main strength of this study, also resulted in some key limitations. First, we were only able to collect data from a small sample which increases the probability of type 2 error: failing to reject the null hypothesis. As this was the case with the direct effects of agreeableness it is worth re-investigating this in other contexts with a larger sample, given that there are studies that have indicated the positive relationship between agreeableness and work engagement [63,64,65].

Moreover, this study concerned a homogeneous sample of working dyads of a hospital in Greece. As a result, our sample cannot be considered as representative of the nursing staff or the working population in general. Future research utilizing similar methods to this study would be needed to clarify whether the present findings can be generalized to the nursing population or even to other occupational contexts.

To build on our findings, future studies can investigate other personality traits which can potentially play a moderating role in the crossover process. For example, along with agreeableness there are studies that indicate that conscientiousness has a positive relationship with empathy e.g., [69,82]. From this perspective it is possible that conscientiousness can also have a positive effect on the crossover process.

The role of agreeableness in crossover processes requires more investigation. It would be important to see if it has a similar effect in dyads in other occupations and work contexts, or in the crossover effects of dual earner couples. It would also be fascinating to examine if agreeableness can enhance crossover effects in bigger work groups and facilitate the emergence of positive affective similarity [83]. Going beyond the overall measure of agreeableness, future studies could add nuance to our understanding of crossover by examining the moderating effect of the sub-dimensions of agreeableness. Whether it is trust, altruism, or any combination of the six sub-dimensions that facilitate the crossover process can have important implications for both theory and practice.

Future studies examining the crossover effect using similar interventions should also improve on the research design used here by collecting daily diary data. Capturing work engagement daily, or even multiple times a day, as well as information regarding how much time dyads spend together, would have allowed a more nuanced representation of how and when crossover happens. Similarly, it would be important to examine the long-term impact of such interventions. Ultimately, we need to know both how long dyads should work together for crossover to happen, but we also need to know for how long the effects of engagement crossover will last after the dyads stop working together.

## 5. Conclusions

Engaged employees tend to find pleasure in their work [84]. They are characterized by high levels of energy and self-efficacy [34] and therefore can benefit an organization and contribute to organizational development and success. This study showed that work engagement crosses over from employees to their colleagues, when they are collaborating closely in a dyad, and that an employee’s work engagement can be enhanced by their own or their partner’s levels of agreeableness. Importantly, the crossover effect was even more likely to occur when both employees within a dyad were characterized by high levels of agreeableness. Finally, the present study indicated that targeted interventions in the working schedule of employees can successfully promote their work engagement.

## Figures and Tables

**Figure 1 ijerph-19-07622-f001:**
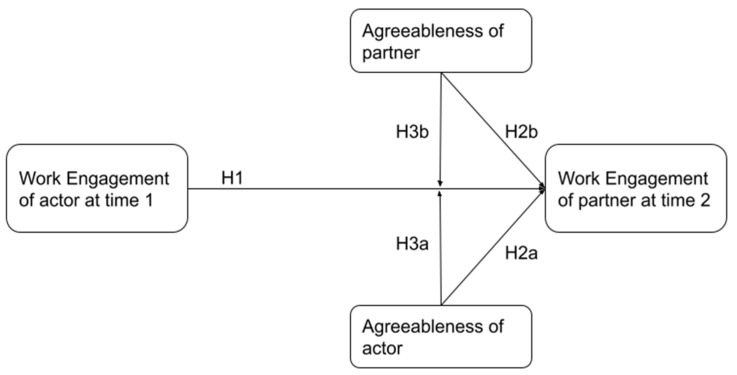
Hypothesized crossover model of work engagement.

**Figure 2 ijerph-19-07622-f002:**
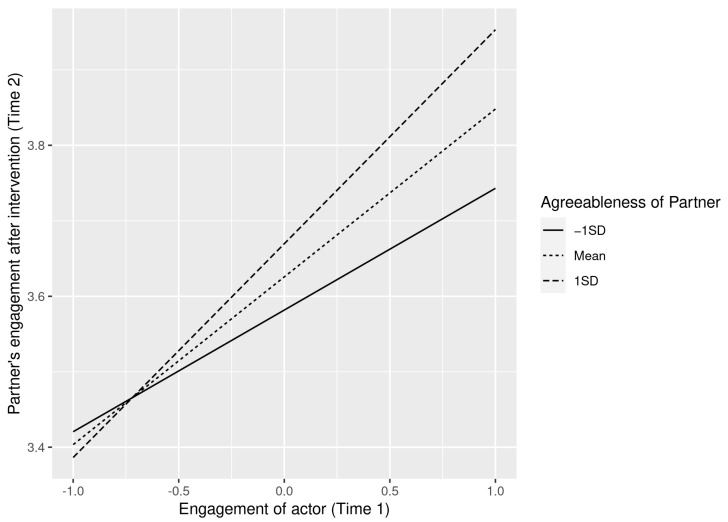
Interaction effect between an actors’ engagement and their partner’s agreeableness on the partners’ engagement.

**Figure 3 ijerph-19-07622-f003:**
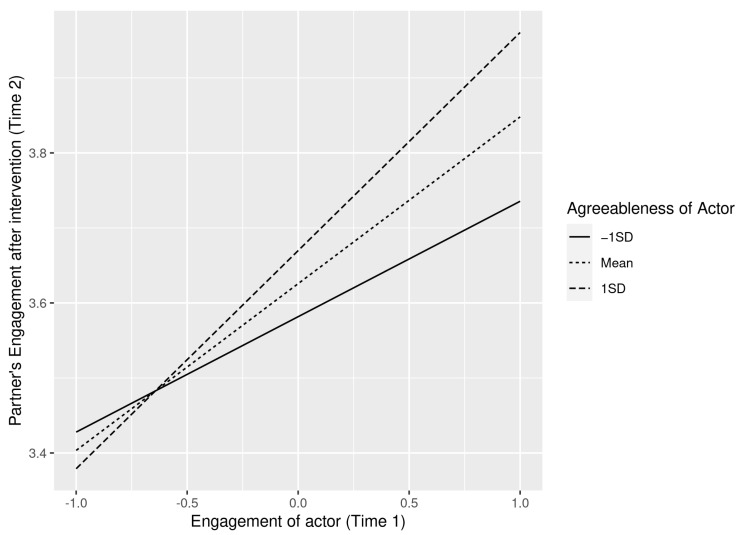
Interaction effect between an actors’ engagement and their agreeableness on their partners’ engagement.

**Table 1 ijerph-19-07622-t001:** Descriptive statistics and correlations for all variables used (N = 74).

		M	SD	1	2	3	4
1	Engagement—Partner T2	3.67	0.95				
2	Engagement—Partner T1	3.50	1.14	0.94			
3	Engagement—Actor T1	3.50	1.14	−0.01	−0.23		
4	Agreeableness—Partner	3.38	0.37	0.21	0.19	−0.08	
5	Agreeableness—Actor	3.38	0.37	0.01	−0.08	0.19	−0.03

r > |0.23| is significant at *p* < 0.05.

**Table 2 ijerph-19-07622-t002:** Effects of agreeableness and actor’s engagement on partner’s engagement.

	Step 1				Step 2				Step 3			
	B	SE	95% CI		B	SE	95% CI		B	SE	95% CI	
			LL	UL			LL	UL			LL	UL
Intercept	3.63 ***	0.03	3.57	3.70	3.63 ***	0.03	3.57	3.70	3.63 ***	0.03	3.57	3.69
Eng. Partner T1	0.95 ***	0.03	0.89	1.01	0.95 ***	0.03	0.89	1.01	0.94 ***	0.03	0.88	1.00
Eng. Actor T1	0.22 ***	0.03	0.16	0.28	0.21 ***	0.03	0.15	0.27	0.22 ***	0.03	0.16	0.28
Agr. Partner					0.04	0.03	−0.02	0.09	0.04	0.03	−0.01	0.10
Agr. Actor					0.05	0.03	−0.01	0.10	0.04	0.03	−0.01	0.10
Eng. Actor T1 * Agr. Partner									0.06 **	0.02	0.01	0.11
Eng. Actor T1 * Agr. Actor									0.07 **	0.02	0.02	0.11
Dyad SD	0.13				0.13				0.15			
Residual SD	0.21				0.21				0.18			
Log Likelihood	1.35				12.62				21.02			
Δχ^2^ (df)	35.95 *** (1)				3.50 (2)				12.28 ** (2)			

N = 74, Dyads = 37. * *p*< 0.05, ** *p*< 0.01, *** *p* < 0.001. CI = confidence interval; LL = lower limit, UL = upper limit. Δχ^2^ is estimated by comparing each model to the one preceding it. Model 1 is compared to a control model with Eng.T1 as control variable.

## Data Availability

Data available upon request from authors.

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
