# Peer review of "Crossover of Work Engagement: The Moderating Role of Agreeableness"

_ijerph, 2022, doi:10.3390/ijerph19137622_

Round 1

Reviewer 1 Report

he publication tackles the interesting topic of infecting engagement with colleagues. The topic is interesting, although it is worth noting that the peak of interest was 15 years ago. The youngest text on the topic in the EBSCO Database is from 2017. One, dating from 2021, refers to work-home relationships. This suggests a certain outdatedness of the issue. This is also expressed in the bibliography, which contains only 8 references no older than 5 years.

In the justification of the contribution to knowledge (lines 58-75) there is a reference to Baker's 2009 text. Pointing to the second aspect of the value of the publication - the replication of other studies (line 64) it is not given which studies are meant. As for the third justification, I find it valid and commendable.

The research strategy seems to me to be interesting and yields interesting results. I regret that the analysis was not undertaken at the level of Agreeblenes factors mentioned in the description of research tools (line 244).

The weakness of embedding in current research is also highlighted by the small volume of the section entitled "Discussion". A real discussion with other studies is missing there. In section 4.1 "Implications for theory and practice" there are references to only 7 publications.

Author Response

Reply to Reviewer 1 comments

Comments:

  1. The publication tackles the interesting topic of infecting engagement with colleagues. The topic is interesting, although it is worth noting that the peak of interest was 15 years ago. The youngest text on the topic in the EBSCO Database is from 2017. One, dating from 2021, refers to work-home relationships. This suggests a certain outdatedness of the issue. This is also expressed in the bibliography, which contains only 8 references no older than 5 years.

Indeed, the peak of interest was around 2010 with a considerable amount of work published. However, as noted by Brough and Westman [1] in their literature review, crossover still gains attention, and additional (and potentially alternative) explanations of the crossover are anticipated and should be introduced. Nevertheless, to address this issue we have embellished our introduction and discussion by citing additional, relevant, and more recent research.

  1. In the justification of the contribution to knowledge (lines 58-75) there is a reference to Baker's 2009 text. Pointing to the second aspect of the value of the publication - the replication of other studies (line 64) it is not given which studies are meant. As for the third justification, I find it valid and commendable.

Thank you for pointing this out. We have now added citations to studies that showed similar results. However, with your comment, we now realize that we may have used the word “replicate” naively as our study is not really a replication of another study. We changed the contribution to say that “…we aim to evaluate and extend previous relevant findings…”.

  1. The research strategy seems to me to be interesting and yields interesting results. I regret that the analysis was not undertaken at the level of Agreeableness factors mentioned in the description of research tools (line 244).

Thank you for the positive feedback regarding our research strategy. We agree that using the agreeableness factors would have yielded more detailed findings and add nuance to our understanding of the role of agreeableness. However, we decided against this because of the interactions in our model. Replacing actor’s agreeableness, and partner’s agreeableness with six dimensions for each would mean that we would have a model with an additional ten predictors and an additional 10 interaction effects. Using all these interactions together could potentially have issues of multi-collinearity because each of these interactions would be with the same variable (engagement of the actor at T1). Similarly, the model would use more degrees of freedom which, given our small sample size, could result in type II errors. We have nevertheless added this in our discussion as a recommendation for future research.

  1. The weakness of embedding in current research is also highlighted by the small volume of the section entitled "Discussion". A real discussion with other studies is missing there. In section 4.1 "Implications for theory and practice" there are references to only 7 publications.

We have now embellished the “Discussion” section by adding more detail to our discussion of the results, as well as more references regarding the implication of the findings and future directions.

References

[1] Brough, P.; Westman, M. Crossover, culture, and dual-earner couples. In The Cambridge handbook of the global work-family interface: Shockley, K.M., Shen, W., Johnson, R.C., Eds.; Cambridge University Press: Cambridge, UK, 2018, pp. 629-645.

Reviewer 2 Report

The abstract section should ideally include a brief summary of the scholarly context to show the study's relevance.

"agreeableness" is a crucial variable in the study, and I think that the first time this term appears in the introduction section, it should be at least briefly described to make it smoother for the general academic reader.

Details in the report of results section must be thoroughly proofread. For example, in lines 332-333, "(step 2: β = 0.04, p < 0.05, 95CI = [-0.02,0.09)" and "(step 2: β = 0.05p < 0.05, 95CI =[-0.01:0.10]) " have problems such as missing spaces and punctuation. I believe that having so many elementary mistakes in a manuscript that is seeking to be published is inappropriate.

The "95CI" in the results report should be a 95% confidence interval, therefore "95% CI" appears more acceptable, and the omission of the "%" sign creates ambiguity. Similarly, the confidence intervals in the table are labelled as (2.5%,97.5%), which is misleading to the reader and should be revised to make it easier to read.

Overall the manuscript should be thoroughly read and proofread as a whole to resolve detailed mistakes.

Author Response

Reply to Reviewer 2 comments

Comments:

  1. The abstract section should ideally include a brief summary of the scholarly context to show the study's relevance.

We have now changed the first half of the abstract to make the scholarly context and aims of the study clearer.

  1. "agreeableness" is a crucial variable in the study, and I think that the first time this term appears in the introduction section, it should be at least briefly described to make it smoother for the general academic reader.

Thank you for highlighting our oversight. We have now added a brief explanation of what agreeableness is immediately after we first mention agreeableness in the introduction. We have kept the more detailed explanation of agreeableness in section 1.3.

  1. Details in the report of results section must be thoroughly proofread. For example, in lines 332-333, "(step 2: β = 0.04, p < 0.05, 95CI = [-0.02,0.09)" and "(step 2: β = 0.05p < 0.05, 95CI =[-0.01:0.10]) " have problems such as missing spaces and punctuation. I believe that having so many elementary mistakes in a manuscript that is seeking to be published is inappropriate.

We have now corrected the typos in the results section and ensured consistency throughout.

  1. The "95CI" in the results report should be a 95% confidence interval, therefore "95% CI" appears more acceptable, and the omission of the "%" sign creates ambiguity. Similarly, the confidence intervals in the table are labeled as (2.5%,97.5%), which is misleading to the reader and should be revised to make it easier to read.

We have changed the 95CI to 95% CI and changed Table 2 according to the format recommended by the APA style manual.

  1. Overall the manuscript should be thoroughly read and proofread as a whole to resolve detailed mistakes.

Thank you for pointing this out. We thoroughly proofread the paper and made corrections throughout. We also had our manuscript proofread by an independent non-academic expert.
